First evidence of bryophyte diaspores in the plumage of transequatorial migrant birds

Lewis Lily R. 1 Lily.Lewis@uconn.edu
Behling Emily 1
Gousse Hannah 1
Qian Emily 1
Elphick Chris S. 1
Lamarre Jean-François 2
Bêty Joël 2
Liebezeit Joe 3
Rozzi Ricardo 4 5
Goffinet Bernard 1
1 Department of Ecology and Evolutionary Biology, University of Connecticut , Storrs, CT , USA
2 Centre d’Études Nordiques et Département de Biologie, Université du Québec à Rimouski , Québec , Canada
3 Audubon Society of Portland , OR , USA
4 Omora Ethnobotanical Park , Puerto Williams, Antarctic Province , Chile
5 Department of Philosophy, University of North Texas , Denton, TX , USA
Traveset Anna
Electronic publication date: 2014 Jun 12
Publication date: 2014
Volume: 2
Electronic Location ID: e424
Received 2014 Apr 8; Accepted 2014 May 20
Copyright: © 2014 Lewis et al.
Copyright year: 2014
Copyright holder: Lewis et al.
License: This is an open access article distributed under the terms of the Creative Commons Attribution License, which permits unrestricted use, distribution, reproduction and adaptation in any medium and for any purpose provided that it is properly attributed. For attribution, the original author(s), title, publication source (PeerJ) and either DOI or URL of the article must be cited.
License URL: https://creativecommons.org/licenses/by/4.0/

Keywords: Bryophyte, Bipolar, Ectozoochory, Long-distance dispersal, Diaspore, Endozoochory, Shorebirds, Sporic, Transequatorial

Funding: University of Connecticut Katie Bu Memorial Fund Switzer Environmental Fellowship National Science Foundation grants DEB 1311405 1212505 Funding for this project comes from the University of Connecticut Katie Bu Memorial Fund awarded to EQ, Switzer Environmental Fellowship awarded to LRL, and National Science Foundation grants DEB 1311405 and 1212505 to LRL and BG, respectively. The funders had no role in study design, data collection and analysis, decision to publish, or preparation of the manuscript.

==============================
Correlations between transequatorial migratory bird routes and bipolar biogeographic disjunctions in bryophytes suggest that disjunctions between northern and southern high latitude regions may result from bird-mediated dispersal; supporting evidence is, however, exclusively circumstantial. Birds disperse plant units (diaspores) internally via ingestion (endozoochory) or externally by the attachment of diaspores to the body (ectozoochory). Endozoochory is known to be the primary means of bird-mediated dispersal for seeds and invertebrates at local, regional, and continental scales. Data supporting the role of bird-mediated endozoochory or ectozoochory in the long distance dispersal of bryophytes remain sparse, however, despite the large number of bryophytes displaying bipolar disjunctions. To determine if transequatorial migrant shorebirds may play a role in the ectozoochory of bryophyte diaspores, we developed a method for screening feathers of wild birds. We provide the first evidence of microscopic bryophyte diaspores, as well as those from non-bryophyte lineages, embedded in the plumage of long distance transequatorial migrant birds captured in their arctic breeding grounds. The number of diaspores recovered suggests that entire migratory populations may be departing their northern breeding grounds laden with potentially viable plant parts and that they could thereby play significant roles in bipolar range expansions of lineages previously ignored in the migrant bird dispersal literature.

Introduction

Climate, geological processes, and long-distance dispersal shape global species distributions. Although wind is the primary vector for long-distance dispersal (LDD) of microscopic diaspores in the Northern (Wilkinson et al., 2012) and Southern (Muñoz et al., 2004) hemispheres, it is an unlikely candidate for explaining bipolar disjunctions (i.e., transequatorial distributions between high latitude areas), which characterize many sporic plants, including at least 60 species of moss (Ochyra, 1992; Ochyra & Buck, 2003; Ochyra, Smith & Bednarek-Ochyra, 2008), 17 liverwort species (Schuster, 1983; Bednarek-Ochyra et al., 2000), and 160 lichen species (Øvstedal & Lewis Smith, 2001). Bipolar disjunctions at the species or infrageneric levels largely originated in the Miocene through the Pleistocene (Wen & Ickert-Bond, 2009; Popp, Mirré & Brochmann, 2011; Fernández-Mendoza & Printzen, 2013) and thus correspond with the continued presence of the Intertropical Convergence Zone (Hyeong et al., 2005), which produces a barrier to wind dispersal across low latitudes. This time frame predates human activities, which in modern times have greatly expanded dispersal opportunities (Wilkinson, 2010). Inferences based on molecular data have largely supported direct LDD across the tropics as the process shaping bipolar disjunctions in plants and lichens (Wirtz, Printzen & Lumbsch, 2008; Wen & Ickert-Bond, 2009; Kreier et al., 2010; Popp, Mirré & Brochmann, 2011; Piñeiro et al., 2012), with only a few examples supporting ancient vicariance (Mao et al., 2012) or stepwise migration across the tropics (Fernández-Mendoza & Printzen, 2013). Bird migration routes between boreal and austral regions provide a direct link between antipodal populations, and hence migratory birds are routinely invoked as the dispersal vectors that account for bipolar disjunctions (Du Rietz, 1940; Wen & Ickert-Bond, 2009; Popp, Mirré & Brochmann, 2011). Evidence of birds dispersing sporic plant or lichen units (i.e., diaspores), as well as those of other lineages, across the tropics, however, is exclusively circumstantial.

Birds may disperse diaspores internally (endozoochory) or externally (ectozoochory; Ridley, 1930; Carlquist, 1974). The feasibility of avian mediated endozoochory in the LDD of aquatic invertebrates and seed plants is well supported by diaspore retention times in captive birds and the survival of ingested diaspores (Proctor, 1968; Figuerola & Green, 2002) as well as the recovery of viable seeds from the dung of wild birds (Bruun, Lundgren & Philipp, 2008). Based on experimentally derived retention times and viability estimates paired with migratory movement data, modeling of potential dispersal distances supports intercontinental-scale movement of aquatic organisms and seeds (Viana et al., 2013). The effects of avian ingestion on the spores of the liverwort Riella was explored by Proctor (1961) who demonstrated that viable spores could be recovered from the dung of domesticated mallard ducks (Anas platyrhynchos) after approximately 30 min. Spores of slime molds have been recovered from the dung of migratory songbirds (Suthers, 1985). Furthermore, bryophyte spores and vegetative diaspores have been shown to withstand ingestion by slugs (Arion vulgaris, A. rufus, and Limax cinereoniger; Boch et al., 2013), or by the spectacled flying fox (Pteropus conspicillatus; Parsons et al., 2007), respectively. Studies on seed plants have shown that the effects of internal passage vary according to bird species and seed (Traveset, Riera & Mas, 2001), but in general, diaspore viability decreases with increasing exposure to a bird’s digestive tract (van Leeuwen et al., 2012).

Alternatively, ectozoochory requires that (1) diaspores become attached to the exterior of a bird prior to migration, and (2) remain on the bird over the course of the journey (Figuerola & Green, 2002). Comparisons of endozoochory and ectozoochory of aquatic invertebrates and seed plants suggest a secondary role of ectozoochory in terms of number and diversity of diaspores dispersed (Brochet et al., 2010; Sánchez et al., 2012; Costa et al., 2014). Several studies, however, suggest that ectozoochory may provide an important means of dispersal for fungi or sporic plants. Viable fungal spores were recovered from the feathers of wild birds up to 45 days after inoculation (Warner & French, 1970) and diaspores of lichen-forming fungi can become attached to the feet of albatross (Bailey & James, 1979). Screening of the coats and hooves of roe deer (Capreolus capreolus), wild boar (Sus scrofa) (Heinken et al., 2001) and domesticated sheep (Pauliuk, Muller & Heinken, 2011) showed that mosses, most notably pleurocarpous mosses, are commonly carried by mammals, despite lacking any specialized means of adhering to them (Sorensen, 1986). Although birds are known to actively transport bryophytes locally for use as nesting material (Osorio-Zuñiga, Fontúrbel & Håkan, 2014), evidence for avian ectozoochory playing a role in the regional or global dispersal of bryophytes is lacking. To test this hypothesis we sought to assess whether the first condition for ectozoochory of bryophytes would be met, namely do potential vectors between high latitude ecosystems harbor bryophyte diaspores.

To evaluate the first condition for ectozoochory by transequatorial migrant birds, we developed a method for microscopically screening feathers for diaspores. We present the results from the screening of 23 individual birds, representing eight transequatorial long-distance migrant species in the order Charadriiformes (shorebirds) captured in their arctic breeding grounds. We provide empirical evidence demonstrating that the first condition for ectozoochorous LDD is met, and with a frequency that suggests that birds are active carriers of diaspores and thus may play a critical role in shaping global species distributions.

Methods

Feather samples were collected from transequatorial migrant birds in their breeding ranges at a site along the Ikpikpuk River, U.S.A. (approximate location: 70.55343°N, 154.69750°W; United States Department of the Interior, permit #23566) and Bylot Island, Nunavut Territory, Canada (approximate locations: 73.15623°N, 79.97065°W; and 72.89216°N, 79.90510°W; Comité de protection des animaux de l’Université du Québec à Rimouski, permit # CPA-42-10-77 - R1) (Fig. 1) between late May and July of 2008 through 2013 by members of the Arctic Shorebird Demographic Network. Target bird species were selected based on migration paths connecting the Northern and Southern Hemispheres, having breeding and wintering ranges in habitats where bryophytes are abundant, and all represent members of the order Charadriiformes. Feather sampling times correspond with the availability of mature bryophyte spores, and preceded the commencement of fall migration. Individual birds were captured using bow nets positioned on nests. When triggered, the bow nets formed a dome over the nest so birds were not immobilized or pressed against the ground, thus lowering the risk that diaspores were picked up due to extraordinary circumstances. Birds were held by grasping the back with clean hands or using nitrile gloves. Feathers were sampled from the base of the breastbone, to minimize invasiveness of feather removal, and based on our prediction that the ventral surface of birds is most likely to make contact with mature bryophyte sporophytes, which are typically erect, or vegetative parts of bryophyte mats. Three to six contour and undercoat feathers were collected using tweezers from each bird and immediately placed into a clean paper envelope. If prebasic molt had begun, older-generation feathers were selected. Following collection, feathers were stored in sealed paper envelopes at room temperature. Contour and undercoat feathers collected from a single individual on the same date were pooled for screening. Feathers from 23 individual birds, representing 8 species, American golden-plover (Pluvialis dominica; n = 11 individuals), red phalarope (Phalaropus fulicarius; n = 3), red-necked phalarope (Phalaropus lobatus; n = 2), ruddy turnstone (Arenaria interpres; n = 1), dunlin (Calidris alpina; n = 1), Baird’s sandpiper (Calidris bairdii; n = 1), white-rumped sandpiper (Calidris fuscicollis; n = 1), and semipalmated sandpiper (Calidris pusilla; n = 3) were screened (Table 1).

Figure 1 Bryophyte diaspores and their vectors.

Three bryophyte diaspores, (A) Sphagnum leaf fragment, (B) liverwort elater, and (C) Bryopsid moss leaf fragment recovered from (D) semipalmated sandpiper (Alaska-6-July-2013), (E) American golden-plover (Canada-30-June-2011) and (F) red phalarope (Alaska-22-June-2013), respectively. 5 µm horizontal scale bars are in the lower right corner of each diaspore image. Maps show Western Hemisphere breeding, migratory, and wintering distributions as well as rare sightings for each bird species (Ridgeley et al., 2012). The migratory and wintering range for red phalaropes (F) overlaps. Bird photo credits: Cameron Rutt.

Table 1 Feather screening results.

Bird species screened, total number of individuals (and feathers) screened, total number of vectors detected (individual bird carrying diaspores) per species, and individual vector identities (reported as location and date of sampling) and number of diaspores recovered per individual. Collection localities are shown in Figs. 1D–1F. Recovered diaspores are shown in Figs. 1A–1C and Figs. 2A–2T. Thirteen diaspores were recovered from red phalarope Alaska-22-June-2013. This bird showed no signs of sickness and did not exhibit any peculiar behaviors. Seven individuals representing three species were found to be vectors out of a total of 23 individuals representing 8 species screened. Small sample sizes likely account for the absence of diaspores in some species.

Bird species	Total individuals screened
(# feathers)	Total # vectors	Total
# diaspores
recovered	Vector ID	# Diaspores per vector	Figures(s)	
American golden-plover	11 (23)	3	6	Canada-30-June-2011	2	Figs. 1B and 2P	
				Canada-7-July-2011	2	Figs. 2I and 2N	
				Canada-13-July-2013	2	Figs. 2B and 2C	
Semipalmated sandpiper	3 (21)	2	3	Alaska-5-July-2013	2	Figs. 2D and 2T	
				Alaska-6-July-2013	1	Figs. 1A	
Red phalarope	3 (14)	2	14	Alaska-22-June-2013	13a	Figs. 1C, 2A, 2F–2H, 2J–2M, 2O and 2Q–2S	
				Alaska-29-June-2013	1	2E	
Red-necked phalarope	2 (8)	0	–	–	–	–	
Ruddy turnstone	1 (3)	0	–	–	–	–	
Dunlin	1 (2)	0	–	–	–	–	
Baird’s sandpiper	1 (1)	0	–	–	–	–	
White-rumped sandpiper	1 (1)	0	–	–	–	–	
TOTAL	23 (73)	7	23	–	–	–	
Notes.

a Red phalarope Alaska-22-June-2013 harbored more diaspores than could be reliably counted, thus 13 representative diaspores were photo-documented.

All screening was performed in a laminar flow hood in which the surface of the work area and all materials were sterilized with 10% bleach, 70% ethanol and 15 min of ultraviolet light exposure. The laminar flow hood was located in a laboratory where bryophytes are not handled. Feathers were placed in a sterile 1.5 mL centrifuge tube with 350 µL of autoclaved distilled water, and vortexed for 5 min at 2,500 rpm to dislodge any particulate matter attached to the feather. The sample was then centrifuged for 5 min at 14,000 g to collect the particulate matter in the base of the microcentrifuge tube. The feather was removed from the tube and placed in a second sterile 1.5 mL microcentrifuge tube for drying. The wash water was then re-centrifuged for 3 min in case the pellet was disturbed when the feather was removed from the tube.

Three hundred µL of supernatant were pipetted off and discarded following the second centrifuging to reduce the sample size for efficient screening. The tube with the remaining 50 µL of the wash water was vortexed to redistribute the pellet, and the solution divided into two 25 µL-samples for microscopic examination. Samples were screened at 40X magnification. The presence of any putative diaspores was recorded and photographed with a Nikon Coolpix E995 camera.

Results & Discussion

We recovered 23 structures representing bryophytes (mosses and liverworts), cyanobacteria, algae and fungi from the small down and contour breast feathers of two semipalmated sandpipers, two red phalaropes, and three American golden-plovers (Table 1; Figs. 1 and 2). Two of the recovered structures are moss leaf fragments. The first is composed of dimorphic cells, with relatively thin chlorophyllose cells alternating with wide hyaline cells, characteristic of Sphagnum leaves (Goffinet, Buck & Shaw, 2009; Fig. 1A). The second has smooth elongate cells, most likely representing a moss from the class Bryopsida, potentially belonging to the family Bryaceae s. lat. or to the the pleurocarpous mosses (superorder Hypnanae; Goffinet, Buck & Shaw, 2009; Fig. 1C). The third bryophyte structure is a liverwort elater, an elongated cell with two helical thickenings (Crandall-Stotler, Stotler & Long, 2009; Fig. 1B). Elaters are dead, hygroscopic structures in the capsule that aid in the dispersal of the spores. The presence of an elater in the plumage highlights that even rare structures, available during limited periods of the year may be picked-up by birds. The recovery of two vegetative diaspores suggests that unspecialized diaspores, which are available consistently throughout the year, may play an important role in ectozoochory of bryophytes.

Figure 2 Diaspores recovered from the feathers of 23 birds.

Twenty of a total of 23 putative diaspores recovered from breast feathers of migratory shorebirds in their breeding ranges. A–G, K, and M are believed to represent green algae or cyanobacteria; H & I meiotic spores, with L representing an immature meiotic product; J, N, and O are multicellular plant fragments; P–T are fungal spores. 5 µm horizontal scale bars are in the lower right corner of each image. Bryophyte diaspores are shown in Fig. 2 with their vectors. Vectors for each diaspore are listed in Table 1.

The behavior of these transequatorial migrant birds in their northern breeding grounds likely promotes their inadvertent acquisition of diaspores. American golden-plovers, semipalmated sandpipers, and red phalaropes all breed in coastal tundra (Tracy, Schamel & Dale, 2002; Hicklin, Gratto-Trevor & Poole, 2010; Johnson & Connors, 2010), where bryophytes are common. Shallow nests are constructed by scraping depressions into the ground with breast, feet and beaks, and are commonly lined with plant materials (Tracy, Schamel & Dale, 2002; Hicklin, Gratto-Trevor & Poole, 2010; Johnson & Connors, 2010). Timing of molt and migratory behavior will affect the likelihood of attached diaspores being dispersed across the birds’ migratory range. Molt in American golden-plovers and semipalmated sandpipers takes place primarily after concluding southward migration, but may occasionally commence before they leave the Arctic (Pyle, 2008; Hicklin, Gratto-Trevor & Poole, 2010; Johnson & Connors, 2010). Red phalaropes replace most body feathers prior to, and flight feathers after, southward migration (Tracy, Schamel & Dale, 2002; Pyle, 2008). American golden-plovers and semipalmated sandpipers make only terrestrial stopovers throughout their migration to South America (Hicklin, Gratto-Trevor & Poole, 2010; Johnson & Connors, 2010), while red phalaropes travel largely over the ocean and spend the non-breeding period offshore (Tracy, Schamel & Dale, 2002). The individuals screened for this study were sampled between June and July (with one exception being a ruddy turnstone sampled on May 31st). Fall migrations in the species found to be vectors commence as early as July for semipalmated sandpipers and red phalaropes, and August for American golden-plovers (Tracy, Schamel & Dale, 2002; Hicklin, Gratto-Trevor & Poole, 2010; Johnson & Connors, 2010), which overlaps with the production of spores by many arctic bryophyte species. Vegetative fragments are constantly available and do not require any temporal correlations between vectors and diaspores. The post-migratory molt and terrestrial destinations of American golden-plovers and semipalmated sandpipers are compatible with the requirements for dispersal across the equator and subsequent establishment of diaspores. The majority of migratory shorebirds with non-breeding grounds in the Southern Hemisphere provide similar opportunities for diaspore dispersal, with the molt typically occurring on the southern non-breeding grounds (Pyle, 2008).

Prior evidence of ectozoochory by birds has shown that this process plays a role in the dispersal of aquatic invertebrates (Dundee, Phillips & Newsom, 1967; Green & Figuerola, 2005; Sánchez et al., 2012), seed plants (Brochet et al., 2010; Costa et al., 2014), and fungi (Warner & French, 1970). Sporic plants commonly exhibit broad intercontinental disjunctions, even across disjunctions that lack wind connectivity. Despite this situation, little work has been done to investigate the possibility of long distance ectozoochory of sporic plants by birds. Our observations provide unequivocal evidence of bryophyte, fungal, and protist diaspores, embedded within the plumage of transequatorial migrant birds, demonstrating that the first condition for bird-mediated ectozoochory is met. Furthermore, our study suggests that vegetative fragments may be significant dispersal units for ectozoochory of bryophytes, supported more broadly by the screening of non-avian vectors (Heinken et al., 2001; Pauliuk, Muller & Heinken, 2011), relative to anemochory whereby spores may be the primary dispersal units. Experimental studies comparing the ability of spores and vegetative fragments to attach and be carried by animal vectors will be necessary to explore this hypothesis further.

The potential of diaspores to establish new populations depends on their sustained viability over the course of dispersal. In bryophytes, resistance of spores to the extreme conditions associated with LDD by wind is a determinant of a given species’ potential for wind mediated range expansions, with continental endemics displaying greater vulnerability during transport than transoceanic disjunct species (van Zanten, 1978). The conditions experienced by diaspores trapped in the plumage of a bird are plausibly less severe than those associated with high atmospheric wind dispersal. Bryophytes are well known for their physiological drought tolerance and the totipotency of their vegetative tissues, which can develop mature plants even after severe grinding (Shaw, 1986), rapid passage through a mammalian digestive tract (Parsons et al., 2007) and even after being frozen under a glacier for 400 years (La Farge, Williams & England, 2013). Future work will explicitly address the second condition for successful ectozoochory by employing culturing techniques to assess viability of recovered diaspores and DNA barcoding for diaspore identification. Based on the general literature supporting the resilience of vegetative bryophyte diaspores, however, resistance to dispersal conditions is unlikely a strong selective force governing the effectiveness of bird-mediated ectozoochory and associated range expansions (Sánchez et al., 2012).

Establishment of a new population is a stochastic event following any mode of dispersal, and may pose the greatest challenge toward range expansions. Considering that 23 diaspores were recovered from seven out of 23 birds sampled, the frequency with which birds may externally transport such structures over time may be sufficiently high to explain bipolar distributions. Extrapolated to entire migratory populations, which range in size in North America from 500,000 individuals for American golden-plovers to 2,260,000 individuals for semipalmated sandpipers (Andres et al., 2012), hundreds of thousands of diaspores may be transported annually across the Equator, dramatically increasing the probability of successful dispersal, establishment and thereby range extension.

Louise Lewis provided access to a laminar flow hood, Richard Lanctot helped in the submission of this project to the Arctic Shorebird Demographic Network, the Arctic Shorebird Demographic Network provided support and collaboration, Margaret Rubega and Matthew Chmielewski provided advice on project development, Susan Hochgraf assisted with preliminary protocol testing, Rafael Medina provided laboratory assistance, and Cameron Rutt contributed bird photos. Andy Green, Casper van Leeuwen, and one anonymous reviewer provided valuable feedback on the initial submission of this work.

Additional Information and Declarations

Competing Interests

Author Contributions

Field Study Permissions

Chris Elphick is an Academic Editor for PeerJ. Joe Liebezeit is an employee of the Audubon Society of Portland and Ricardo Rozzi is an employee of the Omora Ethnobotanical Park. All other authors have no competing interests.

Lily R. Lewis and Bernard Goffinet conceived and designed the experiments, analyzed the data, contributed reagents/materials/analysis tools, wrote the paper, prepared figures and/or tables, reviewed drafts of the paper.

Emily Behling and Emily Qian conceived and designed the experiments, performed the experiments, analyzed the data, wrote the paper, reviewed drafts of the paper.

Hannah Gousse performed the experiments, analyzed the data, wrote the paper, reviewed drafts of the paper.

Chris S. Elphick conceived and designed the experiments, analyzed the data, wrote the paper, reviewed drafts of the paper.

Jean-François Lamarre analyzed the data, contributed reagents/materials/analysis tools, wrote the paper, prepared figures and/or tables, reviewed drafts of the paper.

Joël Bêty and Joe Liebezeit analyzed the data, contributed reagents/materials/analysis tools, wrote the paper, reviewed drafts of the paper.

Ricardo Rozzi analyzed the data, wrote the paper, reviewed drafts of the paper.

The following information was supplied relating to field study approvals (i.e., approving body and any reference numbers):

Canadian samples:

Comité de protection des animaux de l’Université du Québec à Rimouski (CPA-UQAR), permit # CPA-42-10-77 (R1).

United States (Alaska) samples:

United States Department of the Interior, U.S. Geological Survey

Patuxent Wildlife Research Center

Bird Banding Laboratory

Federal Bird Banding Permit, Permit # 23566.

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
