# Peer review of "First evidence of bryophyte diaspores in the plumage of transequatorial migrant birds"

_PeerJ, doi:10.7717/peerj.424_

## Round 0.1 · original submission · Minor Revisions

Your ms has been evaluated by three referees who all agree that your ms will be a valuable contribution to the field of seed dispersal. They all make useful suggestions that I hope will be helpful to you when writting the next version of the ms.

·

Basic reporting

An important paper well worthy of publication after modifications have been made.

The authors use the terms "endochory" and "ectochory". I have never seen this before, and the usual terms are "endozoochory" and "ectozoochory" (OR epizoochory). Either change to these, or cite a source in which your terms have been used before.

Contrary to what the authors state, studies in other organisms suggest many more propagules will be dispersed a long distance by endozoochory, which could potentially account for bipolar distributions, see Brochet et al. 2010, Sánchez et al. 2012 regarding endo vs. ecto.

The suggestion that endozoochory can only disperse propagules over a short distance is also not supported by the broader literature, especially Proctor 1968 and Viana et al. 2013. I think Cruden’s 1966 paper which the authors cite is unreliable as it does not take into account Proctor’s work.

There is a major need for the authors to take into account the historical literature for related taxa, notably Proctor 1961 which shows internal dispersal of bryophytes by waterbirds, and Warner & French 1970 on viability of fungal spores attached to birds. Recent papers well worthy of mention are those by Pauliuk et al. 2011 and Pineiro et al. 2012, as well as the review on ectozoochory by Figuerola & Green 2002.

I agree that wind is unlikely to move spores between hemispheres and hence to explain bipolar distributions, see Wilkinson et al. 2012

line 32, I think the word “if” should be deleted?

Lines 117-119. For plants see also Brochet et al. 2010. But this statement made by the authors is not strictly true, given the evidence also for fungi from Warner & French 1970.

Line 228 Sphagnum is spelt wrong.

Experimental design

How exactly were feathers collected? Was it by shooting birds for example? Please mention the method for bird collection, and consider the risk of whether feathers were contaminated with spores during the collection process, e.g. by falling onto vegetation.

“A few feathers were sampled from each bird”. Please be more specific about what “a few” means, with at least approximate sample size.

In the online resource, I am missing a clear identification of what these 18 diaspores are thought to represent.

The authors made no efforts to demonstrate that the spores they observed were viable. This should be recognized and discussed.

Validity of the findings

The data are important and the study is novel. However, the authors go too far in their suggestion that ectozoochory is more important for endozoochory for these plants, since they do not study endozoochory. I think they should simply suggest that the potential for endozoochory should also be studied in detail.

Additional comments

REFERENCES CITED IN MY REPORT

Brochet, A. L., M. Guillemain, H. Fritz, M. Gauthier-Clerc, and A. J. Green. 2010. Plant dispersal by teal (Anas crecca) in the Camargue: duck guts are more important than their feet. Freshwater Biology 55:1262-1273.

Figuerola, J., and A. J. Green. 2002. Dispersal of aquatic organisms by waterbirds: a review of past research and priorities for future studies. Freshwater Biology 47:483-494.

Proctor, V. W. 1968. Long-distance dispersal of seeds by retention in digestive tract of birds. Science 160:321-322.

Proctor, V. W. 1961. Dispersal of Riella spores by waterfowl. Bryologist 64:58-61.

Sanchez, M. I., F. Hortas, J. Figuerola, and A. J. Green. 2012. Comparing the potential for dispersal via waterbirds of a native and an invasive brine shrimp. Freshwater Biology 57:1896-1903.

Viana, D. S., L. Santamaria, T. C. Michot, and J. Figuerola. 2013. Migratory strategies of waterbirds shape the continental-scale dispersal of aquatic organisms. Ecography 36:430-438.

Warner, G. M., and D. W. French. 1970. Dissemination of Fungi by Migratory Birds - Survival and Recovery of Fungi from Birds. Canadian Journal of Botany 48:907-&.

Wilkinson, D. M., S. Koumoutsaris, E. A. D. Mitchell, and I. Bey. 2012. Modelling the effect of size on the aerial dispersal of microorganisms. Journal of Biogeography 39:89-97.

·

Basic reporting

No comments

Experimental design

No comments

Validity of the findings

Findings are valid, however, the conclusions could be put in slightly better context. An ignorant reader may conclude that the manuscript provides first evidence for "carrying" of spores in the plumage of birds (as stated in the title), while the actual finding is "presence" of spores in the plumage of birds that might migrate over long distances later in the season. Some more specific comments about that below.

Additional comments

This is a well-written manuscript on the presence of spores in the plumage of migratory shorebirds. I only have a few remarks.

Title
The title now suggests the manuscript found “first evidence for carrying spores”, however, if one assumes “carrying” means “transport” there is no further evidence for this in the manuscript. I suggest to change the title to something about the “presence” of propagules on the feathers of birds. Still interesting, but quick readers might otherwise misread and assume the birds carried the spores from or to the breeding grounds.

Abstract

Endochory is more often referred to as endozoochory, similar for ectochory vs ectozoochory, which would make your manuscript more searchable.

As of yet, I am not aware or any plants that specialized their seeds or fruits to facilitate ectozoochory in birds, which is suggested in line 6. While ectozoochory in large mammals has found many adaptations of seeds to attached to fur, in birds, fruits and seed coats are more likely adaptations for endozoochory and not so much to cling to feathers for as far as I know. This would be in contrast to your statement, suggest to rephrase.

Line 13: add a sentence that these diaspores were local and not transported yet, to avoid confusion.


Introduction

Line 12: add a sentence explaining why e.g. humans or wind cannot explain such disjunct distributions. I now assume there is no suitable habitat for these species anywhere between the poles, so if spores land in between on lower latitude areas there is no potential for colonization (even not at high altitudes?).

Line 32: check sentence, “if” seems misplaced

Line 34: maybe you´re interested in reading the following recent papers on endo- and ectozoochory of bryophytes:

Boch, S., Berlinger, M., Fischer, M., Knop, E., Nentwig, W., Türke, M. & Prati, D. (2013) Fern and bryophyte endozoochory by slugs. Oecologia, 172, 817-822.
Pauliuk, F., Muller, J. & Heinken, T. (2011) Bryophyte dispersal by sheep on dry grassland. Nova Hedwigia, 92, 327-341.

Line 35: the part “prior to migration” What about the timing of attachment to birds? You sampled during June and July, when do the birds leave the breeding grounds? See my later comment at line 94.


Methods

Lines 60-65 and Table 1: list all birds that have been checked for the presence of spores. This would give a more complete picture of the frequency of occurrence of spores on feathers, as now from Table 1 the rate of attachment seems higher due to not including the feathers that were checked but did not contain diaspores. This is important for the statement in your last sentence of the abstract.

Line 70 and 71: suggest to check some values. Spinning at 14 g seems very slow, should this be 14000 g? Similar for vortexing at 748 g, is this a conventional lab-bench-vortex?

Line 73: re-centrifuged also at 14000 g?

Results & Discussion

Line 94: discuss the timing effect of carrying spores in the plumage of birds. If such spores are only available during limited times in the year (I am no moss expert) this would have impact on the potential ectozoochory. If the birds carry mosses during June/July but not anymore during August when they start southwards migration, this should be mentioned. If the timing is similar, this could be worth to explicitly add.

Line 116: this paragraph on which birds might be suitable vectors would be interesting to expand into a scheme, or you could rank species for suitability as vectors based on (1) overwintering sites and (2) timing of molt. It is essential that the species flies to suitable habitat (Dunlin overwintering in Mexico may fly a long way but still contribute little to Antarctic biodiversity?) and that the species does not molt before departure. If spores attach to summer plumage, and species molt after the spores have been formed before departure on migration, this attachment may have little ecological relevance. I suggest to discuss this a bit more, perhaps including which bird species are in this scenario the most plausible vectors. This would greatly benefit future research.

Line 120: “sporic plants lack such adaptations”, unless you consider having small spores is an adaptation for dispersal, as small seeds have an advantage in dispersing by zoochory as well as wind.

Line 125-128: although the first part of the sentence is true, the second part is a bit strong. I suggest to refer back to the prerequisites for transport at the end of your introduction for the second part: “showing that…the first part of potential transport requirements is met”. Whether or not this is a “relevant mechanism “ implies a lot of extra speculation, e.g. 1) transport with birds on migration 2) survival of the flight 3) detachment upon arrival 4) arrival in a suitable habitat 5) possibility to establish there 6) that other dispersal mechanisms are not stronger and more important…etc. This you partly discuss further on in the discussion so should maybe not be silently implied already before.

Line 142: more than half of the 21 propagules were found on one bird. Was there something special about this bird? Was it unable to clean its plumage or sick for instance, as often sick birds are full of propagules? It should be mentioned in case the bird was sick, but is also worth mentioning if the bird was perfectly fine.

Good luck with your work!

Casper van Leeuwen

Reviewer 3 ·

Basic reporting

The manuscript adheres to the journal reporting guidelines.

Experimental design

The experimental design is accurately described and appropriate.

Validity of the findings

This manuscript does not set out to statistically test a specific hypothesis but instead it reports a new bit of natural history with potential consequences for the global biogeography of sporic plants. The authors are cautious with the interpretation of their results, as trans-equatorial dispersal is not proven but only suggested based on the presence of diaspores on bird feathers prior to migration.

Additional comments

Major comments
On table 1 the authors indicate the date at which each bird with diaspores was sampled (June-July), but those dates are not put into context regarding the beginning of the migration. The closer the sampling date is to the beginning of the migration, the stronger the support for LDD. I missed a paragraph on the discussion dealing with this issue.

The authors refer to LDD as the only mechanism that can explain the disjoint distribution of sporic plants (lines 122/3). I don’t know enough about the paleo-distribution of these plant groups as to argue otherwise, but I think that the possibility that bipolar distributions might be the result from vicariance from larger distributions from precious glaciations deserves some brief, discussion in this manuscript.

The identity of the 21 diaspores found and their vectors is not comprehensively presented. I think this information should be included into table 1 (see specific comment below).

Minor comments
L25 “Endochory” and “Ectochory” are more commonly referred to as “endozoochory” and “epizoochory” in ecological literature, and I think it would benefit the paper to use the most common nomenclature.
L32 Delete the “if”.
L37/8 Delete “74 breast feathers from”. As feathers collected from the same bird are not independent samples, it would be better to report and discuss the results only based on the number of screened individuals rather than feathers (also in table 1).
L119 Plants with unspecialized and endozoochorous diaspores have also been reported on the feathers of migrating birds, see Costa et al 2013 J. Avian Biol.
L127 Consider using “suggesting” instead of “showing”.
Table1 The table would be a lot more informative if you include a column with the identity of the diaspores found in each vector (i.e. 21 rows). The information on the number of feathers screened can be merged inside brackets into the column “Nº of individuals screened”.
It could also be interesting to include the species and sample size for birds that were screened but had no diaspores.
It’s not clear why you mention that more than 12 diaspores have been found on the Red phalarope, instead of indicating the real number of diaspores found.

---

## Round 0.2 · accepted · Accept

I am glad that you considered all comments given by the three reviewers, which I do think have improved considerably your ms. Your study will surely be a valuable contribution to the field of dispersal ecology.